# Hierarchical Approach to Explaining Poisoned AI Models

## Abstract

This work presents a hierarchical approach to explaining poisoned artificial intelligence (AI) models. The motivation comes from the use of AI models in security and safety critical applications, for instance, the use of AI models for classification of road traffic signs in self-driving cars. Training images of traffic signs can be poisoned by adversaries to encode malicious triggers that change trained AI model prediction from a correct traffic sign to another traffic sign in a presence of such a physically realizable trigger (e.g., sticky note or Instagram filter). We address the lack of AI model explainability by (a) designing utilization measurements of trained AI models and (b) explaining how training data are encoded in AI models based on those measurements at three hierarchical levels. The three levels are defined at graph node (computation unit), subgraph, and graph representations of *poisoned* and *clean* AI models from the TrojAI Challenge.

## 1 Introduction

The *motivation* of this work lies in the lack of interpretability and explainability of artificial intelligence (AI) models in security and safety critical applications. For instance, our lack of understanding of how classes are encoded in AI models for classifying road traffic signs poses a safety threat in self-driving cars because AI models can contain injected triggers causing misclassification Xu et al. (2019).

We introduce the *terminology* used in this paper early on due to a varying usage of published terms in a broad spectrum of theoretical contributions to AI. We will refer to an AI model as a computation graph that (a) is a directed graph representing a math function and (b) consists of subgraphs. A subgraph is a subset of graph vertices (or graph nodes) connected with edges in the parent graph. Graph nodes of a computation graph are computation units (or graph components) that perform linear or non-linear operations on input data, (e.g., convolution, tangent hyperbolic activation function, and maximum value operation). In our work, the names of the AI models (or architectures) are adopted from literature since we are not creating any custom computation graphs. The input and output data at each computation unit are multidimensional arrays denoted as tensors. When an image from a class $c$ flows through a computation graph, each computation unit generates real-valued tensors called class activations. A tensor generated by each input image has dimensions reflecting channels, rows, and columns. In our work, tensor channel values are binarized at zero and denoted as a tensor-state with rows $\times$ columns of tensor-state values.

The *objectives* of this work are (1) to define utilization-based class encodings and AI model fingerprints, (2) to measure class encodings in architectures beyond small models (e.g., LeNet model with 60K parameters) and toy datasets, such as MNIST (Modified NIST dataset with 70K images of size $28 \times 28$ pixels), and (3) to identify encoding patterns (motifs) that discriminate AI models without and with hidden classes (denoted as clean and poisoned AI models). By understanding class encoding patterns, one can additionally benefit from reduced AI model storage and inference computational requirements via more efficient network architecture search Ying et al. (2019) with advanced hardware Justus et al. (2019). Furthermore, one can improve expressiveness of AI model architectures via design Lu et al. (2017) and efficiency measurements Schaub & Hotaling (2020) or one can assist in diagnosing failure modes Bontempelli et al. (2021).

This work addresses the *problems* of (a) designing utlization measurements of trained AI models and (b) explaining how poisoned training data are encoded in AI models based on those measurements.

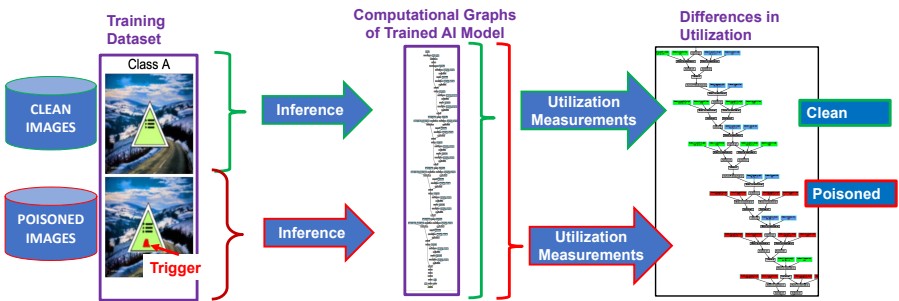

Figure 1: A high-level workflow for identifying utilization patterns in an AI computation graph of ResNet18 architecture for clean and poisoned classes

Table 1: Problems and their complexity challenges for AI models available from TrojAI Challenge, Rounds 1-4 IARPA (2020)

| Problems | Complexity Challenges |
|---|---|
| How to define AI model utilization? | tensor-states in AI models with $\approx 10^{12}$ parameters |
| How to characterize class encoding of each class via utilization of AI model computation units? | $\approx 10^5$ inferences per AI model |
| What AI model computation units are critical for class encodings? | $\approx 10^3$ AI model fingerprints |

Conceptually, utilization of any computation unit is related to a ratio of the number of different outputs (tensor-state values) activated by all training data points over the maximum number of possible outputs by the computation unit. Such utilization-based class encodings are useful as statistical representations of complex trained AI models for (a) classifying a large number of AI models as clean or poisoned, and (b) reducing the search space for understanding class's unique and overlapping patterns. We use a set of tensor-states at each graph node and for each training image as a baseline representation of one trained AI model. With such a baseline representation, one can visually validate correctness of any conclusions derived from utilization-based class encodings for varying class characteristics, application-specific datasets, and AI model architectures.

Figure 1 shows a high-level workflow for identifying discriminating patterns of class encodings in clean and poisoned AI models. The left side in Figure 1 illustrates "Training Dataset" consisting of clean (Class A) and poisoned (Class B) training images with a small red polygon denoted as a trigger (or poison). Training images representing each class are inferenced. During the inference of images from the same class, a vector of utilizations over all graph computation units is recorded and denoted as a class encoding. Differences in class encodings can be visualized by color-coded AI computation graphs to contrast class encodings (e.g., clean and poisoned or clean Class A and Class B - see the right side of Figure 1).

The key *challenges* are enumerated for AI models from TrojAI Challenge IARPA (2020) and summarized in Table 1. The challenges arise as AI architectures are (a) very complex in terms of the number of parameters (from 60K parameters in LeNet model Khan et al. (2020), to common networks having millions and billions of parameters, such as 160 billion reported by Trask et al. Trask et al. (2015), and bleeding-edge networks with trillion-parameters in AI language models Fedus et al. (2021)), (b) very heterogeneous in terms of types of computation units in computational AI graphs, and (c) high dimensional in terms of data tensors generated by AI graph computation units.

The underlying *assumption* of our approach is that the tensor-state statistics at each graph node reveal a presence or absence of hidden classes (triggers or backdoor attacks). The assumption is supported by a successful defense against backdoor attacks by graph pruning Liu et al. (2018) and the fact that encoding a hidden class will increase the utilization of some graph nodes as measured by tensor-state statistics. Furthermore, although symbolic representations of subgraphs are still under investigation

Olah et al. (2020), we also assume that the utilization-based characterization of subgraphs may have a relationship with symbolic descriptions of image parts (e.g., subgraphs encode a traffic sign shape) and hence presence or absence of trojans can be detected by finding patterns in utilization-based color-coded graphs and subgraphs.

The main *novelties* of this work are in the definition, measurement design, and pattern searching in utilization-based clean and poisoned class encodings. The main *contributions* are in utilization measurement placements for a variety of AI architectures, and in explainable clean and poisoned AI models at the granularity levels of AI model graphs, subgraphs and tensor-states. Our work leveraged interactive Trojan and AI efficiency simulations enabled by the Neural Network Calculator tool Bajcsy et al. (2021) and web-accessible AI models generated for the TrojAI Challenge computer vision rounds IARPA (2020).

## 2 RELATED WORK

The problem of explainable AI is very broad and the term *explainable* is still debated in philosophical texts Gilpin et al. (2018) ("What is an Explanation?"). A comprehensive survey of explainable AI has been published by Arrieta et al. Barredo Arrieta et al. (2020) and extensive teaching materials have been made available by Lakkaraju et al. Lakkaraju et al. (2020). Our approach can be related to "Explanation of Deep Network Representation" (roles of layers, individual units, and representation vectors) according to the Deep Learning-specific taxonomy presented in Barredo Arrieta et al. (2020), Fig. 11. Our utilization-based approach is inspired by exploring relationships between biological neural circuits and AI model computation graphs as discussed in Olah et al. (2020). Next, the related work is presented with respect to the three formulated problems.

Our work on *defining utilization* is related to the past work on measuring neural network efficiency Schaub & Hotaling (2020), Bajcsy et al. (2021), which is rooted in neuroscience and information theory. In the work of Schaub and Hotaling Schaub & Hotaling (2020), neural efficiency and artificial intelligence quotient (aIQ) are used to balance neural network performance and neural network efficiency while inspired by the neuroscience studies relating efficiency of solving Tetris task and brain metabolism during the task execution Haier et al. (1992). In the work Bajcsy et al. (2021), an online simulation framework is used to simulate efficiencies of small-size neural networks with a variety of features derived from two-dimensional (2D) dot pattern data. In contrast to the previous work Schaub & Hotaling (2020), Bajcsy et al. (2021), our theoretical framework defines and reasons about class encodings, AI model fingerprints, and metrics for finding class encoding patterns for much more complex AI models and training datasets.

Following the categorization in the survey on interpreting inner structures of AI models Räuker et al. (2022), the *utilization measurements* can be related to concept vectors whose goal is to associate directions in latent space with meaningful concepts. In the works Fong & Vedaldi (2018)(Network 2 Vector) and Bau et al. (2017) (Network Dissection), the distribution of activation maps at each convolutional unit as inputs pass through is used to determine a threshold. Threshold-based segmented activation maps are compared across concepts. In contrast to the previous work Fong & Vedaldi (2018),Bau et al. (2017), our utilization measurements are computed at all computation units in an AI model, the activation maps are binarized at zero, and statistics are computed over a distribution of tensor-states (including the binarized activation maps from convolutional units). Our approach does not use any inserted modules like in concept whitening Chen et al. (2020) to align the latent space with concepts. Furthermore, our approach does not project class activation maps to create saliency maps Selvaraju et al. (2019), Adebayo et al. (2018) in the input spatial domain, but, rather, it analyzes class activations in the tensor-state space.

Finally, following the categorization of approaches to understanding community (group or cluster) structure in AI models presented in Watanabe et al. (2018), our overarching approach to *finding class encoding patterns* falls into the category "Analysis of trained layered neural networks" and combines two subcategories: analysis of unit outputs and their mutual relationships and analysis of the influence on neural network inference by data. Overall, our approach can be related to modular partitioning Hod et al. (2021), Filan et al. (2021), and unsupervised disentanglement of a learned representation Locatello et al. (2018), Locatello et al. (2020). In contrast to Hod et al. (2021), Filan et al. (2021), our clustering of computation units does not use "strong" and "weak" structural undirected connectivity of neurons as in spectral clustering, but, rather, repetitive co-occurrence patterns of utilization values

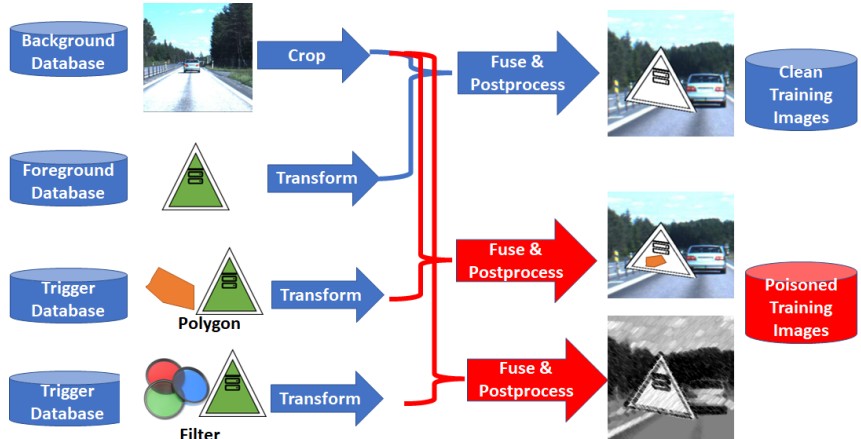

Figure 2: The process of creating training data with traffic signs. An example of a simulated triangular traffic sign and polygon or filter types of triggers.

in connected computation units. While we tacitly assume that high-dimensional data can be explained by lower dimensional semantically meaningful latent variables as in Locatello et al. (2018), Locatello et al. (2020), we do not attempt to fully automate finding subgraphs (i.e., a human is always in the loop) to follow the published conclusions. In addition, we do not aim at fully partitioning all AI model computation graphs into poly-semantic and mono-semantic subgraphs based on the poly-semantic and mono-semantic classification of neurons in subgraphs according to Olah et al. (2020) and Räuker et al. (2022).

## 3  METHODS

In this section, utilization-based class encodings are defined by addressing the three key challenges listed in Section 1: (1) AI graph size and connectivity complexity, (2) component (graph node) heterogeneity, and (3) tensor dimensionality and real-value variability. The utilization measurements of class encodings are defined by introducing tensor-states measured at the output of each component in AI computation graphs as training data points pass through the AI graph. The process of creating clean and poisoned training datasets is described next.

**Creation of clean and poisoned training datasets:** The training images for each class in TrojAI challenge (Rounds 1-4) are created according to Figure 2 by fusing and post-processing foreground and background images. Images of foreground traffic signs are constructed from images of real and simulated traffic signs. The background images are retrieved from existing road and city video sequences (e.g., citiscapes Cordts et al. (2016), KITTI 360 by Karlsruhe Institute of Technology and Toyota Technological Institute at Chicago Menze & Geiger (2015), and others Bruno et al. (2022)). A variety of images per traffic sign class is accomplished by changing parameters of crop, transformation, fusion, and post-processing operations as shown in Figure 2.

**Utilization-based Class Encoding:** We start with the following definitions.

*Clean and Poisoned AI models:* Let $F_a : \mathbb{R}^m \to \{1, ..., C\}$ refer to a trained AI model with architecture $a$ that classifies two-dimensional $m$-variate images into one of $C$ classes. When $F_a$ is clean (denoted as $F_a^{\square}$), $F_a$ achieves a high classification accuracy over input images $\vec{x}_i \in \mathbb{R}^m; i \in \{1, ..., M\}$ where $M$ is the number of pixels. When a clean $F_a$ is poisoned by a trigger (denoted as $F_a^{\blacksquare}$), there exists a function $g : \mathbb{R}^m \to \mathbb{R}^m$ applied to input images from a source trigger class $c_s$, such that $F_a(g(\vec{x}_i)) = c_t$, where $c_t$ is the target trigger class and $c_t \neq c_s$. Examples of clean images from source class, poisoned images from source class, and clean images from target class are shown in Figure 2. For a pair of trained clean and poisoned AI models, labels for source class $c_s$ and target class $c_t$ are predicted with high accuracies according to the four equations below:

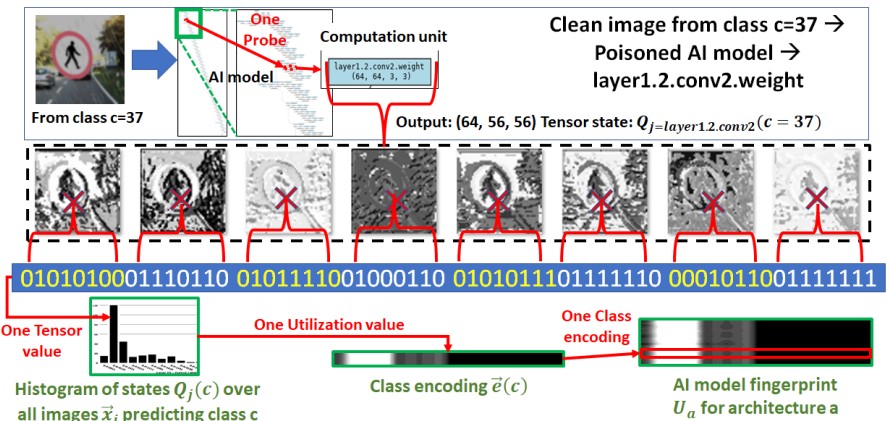

Figure 3: An example of measuring values of one tensor-state by placing a probe at one node of the ResNet101 computation graph (node layer1.2.conv.weight) and recording the values for one input image. The values of the tensor-state contribute to a histogram of states used for deriving one utilization value per node in the class c encoding that contributes to an AI model fingerprint as illustrated by red arrows).

$$F_a^\square(\vec{x}) = c_s \ \text{ and } \ F_a^\square(g(\vec{x})) = c_s \tag{1}$$

$$F_a^\blacksquare(\vec{x}) = c_s \ \text{ and } \ F_a^\blacksquare(g(\vec{x})) = c_t \tag{2}$$

*AI computation graph:* A computation graph of a trained AI model $F_a$ is denoted by $G_a = \{V, E\}$ where $V = \{v_1, v_2, ..., v_{n(a)}\}$ are the $n(a)$ computation units (or graph nodes or graph components) and $E \subseteq V \times V$ are the edges. The unidirectional edges of a graph $G_a$ are described by an adjacency matrix $A \in \{0, 1\}^{n(a) \times n(a)}$ with $A_{ij} = 1$ for all connected nodes $v_i$ and $v_j$, and $A_{ij} = 0$ for all other node pairs.

*Tensor-state:* Each input image $\vec{x}_i$ passes through $G_a$ populated with trained coefficients. The input generates a tensor of output values at each computation unit (i.e., an activation map) $v_j : \mathbb{R}^{D_j^{In}} \to \mathbb{R}^{D_j^{Out}}$, where $D_j^{In}$ and $D_j^{Out}$ are the input and output dimensions of data at the computation unit $j$. The output values are binarized by zero value thresholding to form a tensor-state $s_j(\vec{x}_i) = b(v_j(\vec{x}_i)) \in \{0, 1\}^{D_j^{Out}}; b : \mathbb{R}^{D_j^{Out}} \to \{0, 1\}^{D_j^{Out}}$. We refer to the graph location of $v_j$ at which the output values are measured as a probe location. Figure 3 illustrates one tensor-state value for a specific ResNet101 computation graph, its specific graph node named layer1.2.conv2.weight, and one image from a predicted class $c = 37$. The example tensor-state $(64, 56, 56)$ is visualized as a set of 8 images with dimensions $56 \times 56$ pixels, and the 64 bits (binarized outputs) are represented as 8 bytes.

*Tensor-state Distribution:* Given a set of measured tensor-states $\{s_j(\vec{x}_i)\}$ at a computation unit $v_j$ for which $F_a(\vec{x}_i) = c$, let us denote $Q_j(c) = \{q_{ij}(c)\}_{i=1}^{n_j}$ to be a discrete probability distribution function (PDF) over all tensor-state values, where $n_j = 2^{D_j^{Out}}$ is the maximum number of available tensor-state values at the $j$-th computation unit $v_j$. The value of $q_{ij}(c)$ is the sum of counts of unique tensor-state values $count_{ij}$ invoked by all images $i$ ($\bigvee i \to s_j(\vec{x}_i)$) and normalized by the maximum number of available tensor values $n_j$. Figure 3 (bottom left) shows the histogram values $count_{ij}$ computed from 5 366 576 unique tensor-state values over all 2 500 training images of *STOP pedestrian crossing* traffic signs. Based on the tensor-state dimensions $(64, 56, 56)$, one can establish the maximum number of predicted classes for such a node to be $C_{layer1.2.conv2}^{MAX} = \frac{2^{64}}{56 * 56 * 2500} \approx 2.35 * 10^{12}$; a terascale count of traffic sign classes.

*Reference tensor-state distribution:* For a class-balanced training dataset with similar class complexities, let us refer to $P_j = \{p_{ij}\}_{i=1}^{n_j}$ as the uniform (reference) PDF over all states; $p_{ij} = \frac{1}{2^{D_j^{Out}}}$. The probabilities $p_{ij}$ are associated with each state (index $i$) and each computation unit (index $j$) for each class $c$.

*Utilization:* We can compute a scalar utilization value $\eta_j(c)$ for each class label $c$ and a computation unit $v_j$ from the count of measured states $q_{ij}(c)$ and the state distribution $Q_j(c)$ based on Equations 3-5. Equation 3 defines utilization $\eta_j^{state}$ based on a deterministic view of states. In contrast, Equations 3 and 4 define utilizations $\eta_j^H$ and $\eta_j^{KLDiv}$ based on a probabilistic view of states by computing entropy $H(Q_j)$ of a state distribution normalized by maximum entropy $H_j^{max}$ or reference distribution $P_j$. The three utilization definitions yield value ranges $\eta_j^{state} \in [0,1]$, $\eta_j^H \in [0,1]$, and $\eta_j^{KLDiv} \in [0,\infty]$ per computation unit with an index $j$. For increasing utilization, the state- and entropy-based measurements will increase while the Kullback–Leibler(KL) Divergence-based measurement will decrease since it measures non-utilization (or a deviation from the reference uniform distribution of tensor-states across all predicted classes). The KL Divergence-based measurement assumes that the maximum number of available states $n_j$ is uniformly divided across all predicted classes (i.e., class encodings consume an equal number of available tensor-states).

$$\eta_j^{state} = \sum_{i=1}^{n_j} \frac{count_{ij}}{n_j} = \sum_{i=1}^{n_j} q_{ij} \leq 1 \tag{3}$$

$$\eta_j^H = \frac{H(Q_j)}{H_j^{max}} = \frac{-\sum_{i=1}^{n_j}(q_{ij} * \log_2 q_{ij})}{\log_2 n_j} \tag{4}$$

$$\eta_j^{KLDiv} = D_{KL}(Q_j \parallel P_j) = \sum_{i=1}^{n_j}(q_{ij} * \log_2 \frac{q_{ij}}{p_{ij}}) \tag{5}$$

The vector of utilization values for all AI computation units $j \in \{1, ..., n(a)\}$ is referred to as *a class encoding $\vec{e}(c)$ for the class $c$.* The vector of utilization values from all classes $c \in \{1, ..., C\}$ is referred to as *a probe encoding $\vec{r}(j)$ for the computation unit $j$.* A set of class encodings for $c \in \{1, ..., C\}$ ordered by the class label is denoted as *an AI model utilization fingerprint $\mathbf{U_a} = \{\vec{e}(c = 1), ..., \vec{e}(c = C)\}$.* In terms of utilization properties, the values are nondecreasing for increasing number of training data, number of predicted classes, decreasing AI model capacity.

**Class Encoding Measurements:** Following the theoretical definition, the utilization measurement workflow steps are shown in Figure 3. The workflow starts with placing multiple measurement probes to collect the activation maps and follows the sequence of steps in Figure 3: record tensor-states, compute a histogram of tensor-states, derive class encoding for one class, and form an AI model utilization fingerprint. The placement of a measurement probe is after each computation unit.

The measurement involves building state histograms, computing the utilization values according to Equations 3 - 5 per computation unit of AI computation graph, and repeating the calculations over hundreds of computation units per graph while evaluating hundreds of thousands of images per AI model and thousands of trained AI models. We approached the computational challenges by

- reducing the number of training images per class and building an extrapolation model,
- analyzing the AI model architecture designs to limit the number of probes, and
- modifying the KL Divergence computation to reduce computations.

Utilization values are measured by evaluating clean and poisoned trained AI models by clean and poisoned training images.

**Finding Patterns in Utilization-based Class Encoding:** We search for patterns based on the following measurements and metrics at each granularity level:

- Graph node: spatial overlaps of semantically meaningful image regions with tensor-state values (e.g., common blue sky in all class-specific images versus class-unique traffic sign symbols),
- Subgraph: partial similarities of multiple class encodings in the same AI model (e.g., encoding of traffic sign classes utilizing similar and dissimilar AI model computation subgraphs), and
- Graph: similarity of AI model utilization fingerprints in collections of AI models (e.g., common utilization of multi-class encodings in multiple AI model architectures).

Table 2: Four trained models with the following parameters: architecture $a = \text{ResNet101}$, number of predicted classes $C = 17$, number of trojans $g_i(\vec{x})$ per AI model $\{0, 1, 2, 2\}$, and trigger functions defined below.

| Model ID | Model Type | Trigger 0 | Trigger 1 |
|----------|-----------|-----------|-----------|
| 142 | Clean | $g_0(\vec{x}) = \vec{x}$ | $g_1(\vec{x}) = \vec{x}$ |
| 235 | Poisoned | $g_0(\vec{x}) = \text{Kelvin filter}$ | $g_1(\vec{x}) = \vec{x}$ |
| 150 | Poisoned | $g_0(\vec{x}) = \text{Gotham filter}$ | $g_1(\vec{x}) = \text{Lomo filter}$ |
| 250 | Poisoned | $g_0(\vec{x}) = 9\text{-sided polygon}$ of color $[200, 0, 0]$ | $g_1(\vec{x}) = 4\text{-sided polygon}$ of color $[0, 200, 200]$ |

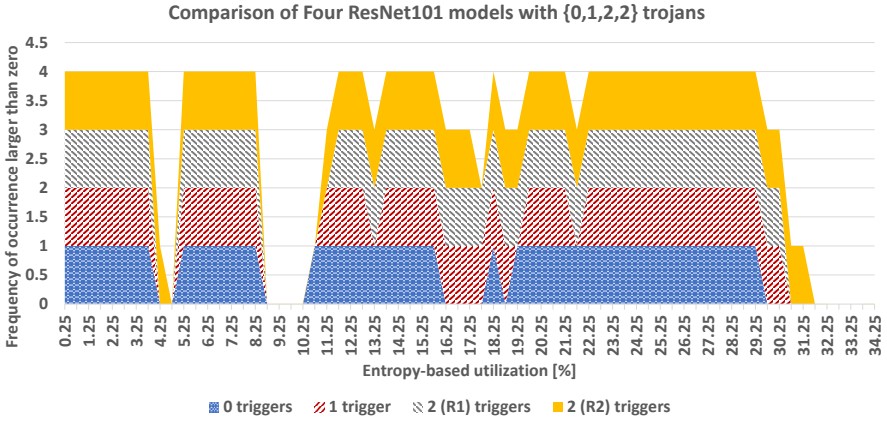

Figure 4: Comparison of four ResNet101 models with zero, one, and two triggers (two replicates denoted as R1 and R2).

## 4 EXPERIMENTAL RESULTS

**Finding Patterns by Comparing Clean and Poisoned Classes:** We started with pattern detections in computation graphs first, next in subgraphs, and then in graph nodes. Our experiments are motivated by (a) evaluating our hierarchical utilization-based approach to classifying a large number of AI models and (b) understanding and validating the use of utilization measurements for this classification task at the tensor-state (micro) levels.

**Patterns detected in computation graphs:** We illustrate the utilization patterns in class encodings for four trained models in Round 4 holdout dataset of TrojAI Challenge IARPA (2020) with the parameters summarized in Table 2. The four AI models are trained with different traffic signs, assigned randomly to 17 classes, and placed on top of randomly chosen backgrounds from cityscapes, kitti road, and kitti city image collections, and, therefore, the fingerprints cannot be compared by element-to-element.

All four models have approximately the same distribution of utilization values over all encoded traffic classes. However, as can be seen in Figure 4, there are utilization values in ranges $[16.0, 18.0] \cup [18.5, 19.0]$ and $[29.5, 31.5]$ that are present in the poisoned models but are missing in the clean model. The utilization values in $[16.0, 18.0] \cup [18.5, 19.0]$ are measured at the computation units labeled as maxpool, conv1, bn1, and ReLU (maximum pooling, convolution, batch normalization, and rectified linear unit). The utilization values in $[29.5, 31.5]$ come from layer1.2.conv2 and layer1.2.bn2 in all poisoned models. In addition, the values in $[29.5, 31.5]$ are also measured in AI models poisoned with polygon triggers at the computation units labeled as layer1.1.conv2 and layer1.1.bn2. Based on this granularity-level analysis, one can focus on the identified subset of computation units to explain the clean versus poisoned class encodings.

**Patterns encoded for clean versus poisoned classes in computation subgraphs:** Figure 5 shows the comparison of clean class encoding $c = 25$ (left) and two replicate class encodings of $c = 25$ with Kelvin Instagram filter as a trigger (middle and right) in the ResNet101 architecture. Based on

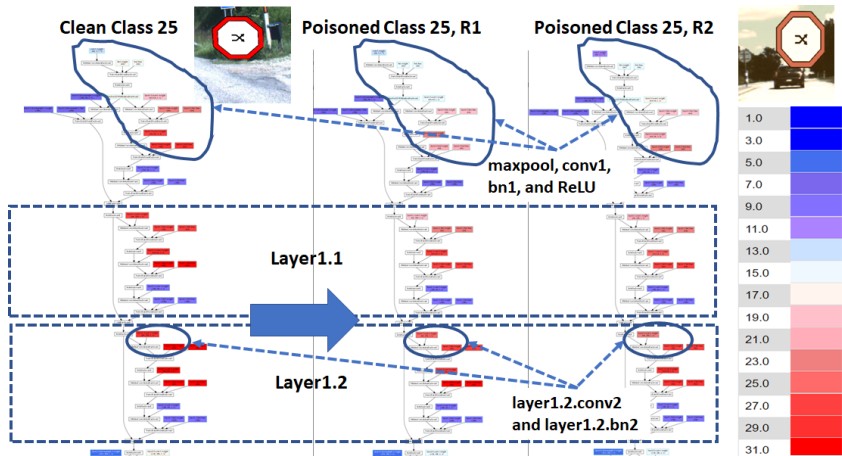

Figure 5: Comparison of a clean class encoding evaluated with clean images (left) and of poisoned class encodings in two trained replicate AI models (ResNet101 architecture) evaluated with poisoned images (middle and right). The circles show the variability of utilization in the initial graph nodes and layer1.0 in two poisoned class encodings. The rectangles show the utilization pattern change between clean and poisoned class encodings.

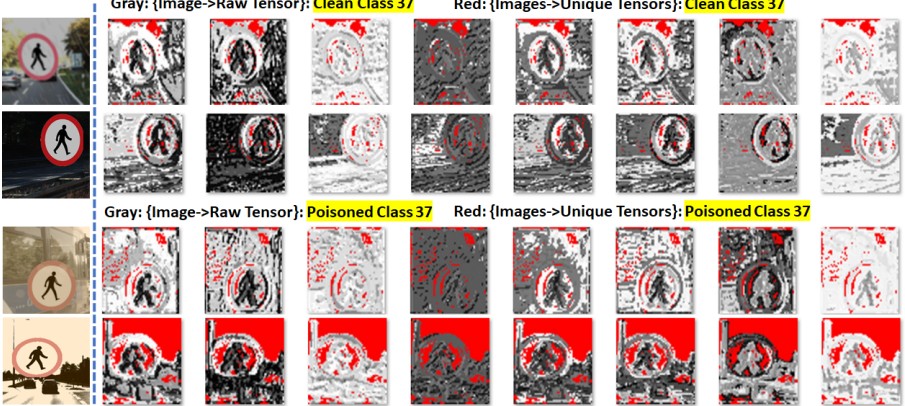

Figure 6: Visualization of tensor-state values in red for two sample clean (top two rows) and poisoned (bottom two rows) images from the same class in layer1.2.conv2 of ResNet101 that occur more than 100 times in 2500 clean images (top two rows) or in 2500 poisoned images (bottom two rows).

the AI model fingerprint analyses in Section 4, Instagram filters and polygons as triggers present themselves in the initial maxpool, conv1, bn1, and ReLU computation units. Varying utilization (different from the clean class encoding) can be observed in Figure 5 with the circles enclosing maxpool2d, ReLU, conv1.weight, layer1.0.conv1.weight, layer1.0.conv2.weight, layer1.0.bn2.weight, and layer1.0.bn2.bias. The color coding goes from dark blue to dark red or from 1 % to 31 % of entropy-based utilization (see the color legend in Figure 5).

Regarding the subgraph pattern 1 shown in Figure 5, the trigger of Kelvin Instagram filter type breaks the pattern between layer1.1 and layer1.2 as highlighted with two dash-line rectangles in Figure 5. Since the Kelvin Instagram filter reduces the color spectrum to the earth tones of green, brown, and orange, this will reduce the number of unique tensor-states and, hence, reduce the utilization of some computation units.

**Patterns detected in computation units:** We compared the tensor-states characterizing clean and poisoned classes in Figures 6 and 7. The comparison of clean and poisoned classes is shown for the same *STOP pedestrian crossing road* sign with or without applied Kelvin Instagram filter as a

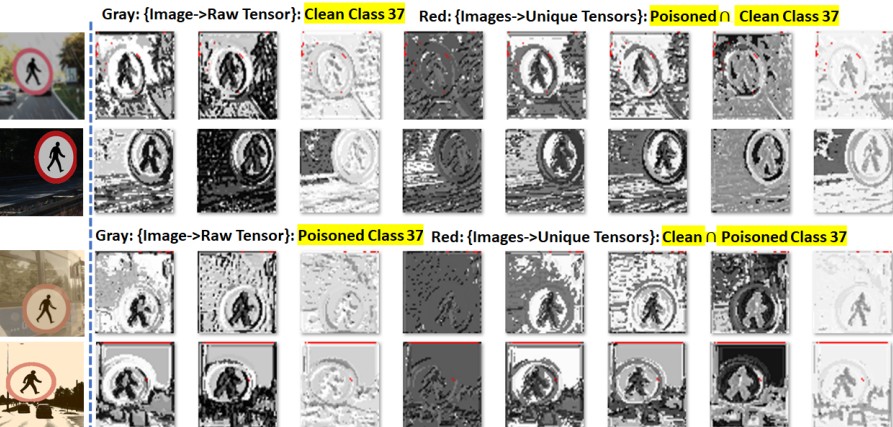

Figure 7: Visualization of tensor-state values in red for two sample clean (top two rows) and poisoned (bottom two rows) images from the same class (*STOP pedestrian crossing road* signs) in layer1.2.conv2 of ResNet101 that occur more than 100 times in both 2500 clean and 2500 poisoned images.

trigger. Figure 6 (top two rows) illustrates that the common tensor-state values within a clean class correspond to sky, parts of a road without shadows, and several pixel clusters inside the traffic sign. After applying the Kelvin Instagram filter, Figure 6 (bottom two rows), the common tensor-state values within a poisoned class are dominated by the earth tones of green, brown, and orange, which leads to a merger of semantically distinct regions, such as sky, parts of the road, and interior of the traffic sign. The image in the lowest row of Figure 6 has a significant number of red pixels suggesting that it consists of many features common across all poisoned images.

The objective of Figure 7 is to visualize with red pixels any common tensor-states across clean and poisoned classes. All four rows in Figure 7 show almost no red pixels except from a few pixels from the yellowish tree and from a red rim of the traffic sign in the top row left image. Since the Kelvin Instagram filter affects every pixel in a training image, the overlap of high-frequency tensor-state values between clean and poisoned images is only 35 tensor-state values and almost none in the area of the *STOP pedestrian crossing* traffic sign. In other words, although perceptually the areas of clean and poisoned traffic signs are very similar, the features characterizing each class as generated by the computation unit layer1.2.conv2.weight are completely different. Furthermore, since the Kelvin Instagram filter blurs pixel values but makes their color more similar to each other, there are less unique tensor-state values in poisoned images than in clean images.

## 5 SUMMARY

We have introduced the concept of AI model utilization for the purpose of explaining clean and poisoned AI models at graph, subgraph, and tensor-state granularity levels. We defined a mathematical framework for computing three deterministic and statistical AI model utilization metrics. We benchmarked the computational cost of inferencing M = 2500 images on NVIDIA Titan RTX with n(a) = 286 probes in a =ResNet101. The computation took on average 24.46 minutes while the memory consumption reached up to 140.6 GB for one model from the TrojAI Challenge. Furthermore, we implemented a suite of tools for measuring utilizations of each computation unit in a computation graph and visualized the utilization measurements as matrices (AI model fingerprints), color-coded graphs, and a sequence of images representing a multidimensional array. Specifically, we explained the utilization-based class encodings for clean and poisoned classes from the TrojAI Challenge (Rounds 1-4) IARPA (2020). We concluded that while clean and poisoned images can clearly be classified into the same semantic traffic sign category, a poisoned AI model would have completely independent tensor-states for clean versus poisoned traffic sign images (see Figure 6 versus Figure 7). The limitation of the current work is in visual analyses of subgraph patterns and unique graph nodes for clean and poisoned AI models which is the topic of our future work.

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
