# OpenReview forum: "Hierarchical Approach to Explaining Poisoned AI Models"
_ICLR.cc/2024/Conference — Submitted to ICLR 2024_

### Official Review · Reviewer_VtiR · 2023-10-18

**Soundness:** 2 fair
**Presentation:** 1 poor
**Contribution:** 2 fair
**Rating:** 3
**Confidence:** 2

**Summary:**

This paper presents a hierarchical approach to explaining poisoned artificial intelligence (AI) models. In particular, the authors design utilization measurements of trained AI models (in the form of computational graphs) and explain how training data are encoded in AI models
based on those measurements at three hierarchical levels (graph node, subgraph, and graph). The author evaluates their method by showing their definition (at three different levels) can result in different patterns and thus be identified for clean and poisoned classes.

**Strengths:**

efficiently differentiating clean models from poisoned models seems a potential interesting problem.

**Weaknesses:**

1.experiments are not sufficient

(1) limited qualitative examples are given, no quantitative evaluation

(2) no baseline is compared with

2.writing needs significant improvement

(1) flow needs improvement

Many parts of the paper are not connected well. It’s better to add connection and summary for each section/subsection to inform the reader what to expect and the connections among paragraphs.

(2) content can be better organized

For example, it is very unusual to put computational setup (which usually appears in the experiment section) in the summary section.

**Questions:**

-Quantitative evaluation and baselines?

---

> ### Author Response · Authors · 2023-11-17
> **response to reviewer VtiR**
>
> Weaknesses:
>
> 1.experiments are not sufficient
>
> (1) limited qualitative examples are given, no quantitative evaluation
>
> (2) no baseline is compared with
>
> Response: The page limit did not allow us to include more qualitative examples, We would appreciate if the reviewer would suggest a baseline method to compare our utilization based hierarchical approach to explaining trojan encodings. We are only aware of the work by Chris Olah - https://80000hours.org/podcast/episodes/chris-olah-interpretability-research/. However, it is pure exploratory research with many hypotheses.
>
> 2.writing needs significant improvement
>
> (1) flow needs improvement
> Many parts of the paper are not connected well. It’s better to add connection and summary for each section/subsection to inform the reader what to expect and the connections among paragraphs.
>
> (2) content can be better organized
> For example, it is very unusual to put computational setup (which usually appears in the experiment section) in the summary section.
>
> Response: Thank you for the recommendation to add connections and summary for each section/subsection. We will move the computational setup from the summary section to the experimental section per recommendation as well.

---

> > ### Comment · Reviewer_VtiR · 2023-11-23
> > **Thank you for the response!**
> >
> > Thank you for the response! However, my main concern (limited qualitative examples are given, no quantitative evaluation) has not been addressed. Also, no updated pdf has been provided to address my writing concern.

---

### Official Review · Reviewer_f9Xz · 2023-10-28

**Soundness:** 2 fair
**Presentation:** 2 fair
**Contribution:** 2 fair
**Rating:** 3
**Confidence:** 4

**Summary:**

This paper focuses on the explanation of AI models, especially the models under backdoor attack. Considering the different activation states of neurons between clean and poisoned inputs in clean and poisoned models, the authors define the utilization of neurons to represent their activation patterns. Then, they represent the activation patterns of models w.r.t. each specific class based on the utilization of neurons. By comparing the activation patterns in clean and poisoned classes and models, the authors identify the spatial location where the attack takes effect.

**Strengths:**

- This paper is well-motivated, i.e., the authors attempt to explain the poisoned classes and models in the backdoor attack, which is crucial to the safety of AI applications.
- The authors clearly introduce the motivation, objectives, problems to address, and challenges of this study in the introduction.

**Weaknesses:**

- There have been some similar metrics to the proposed method in this paper, and I think the method in this paper has no significant superiority or advantages over previous methods. For example, Ma et al., (2018) and Bai et al., (2021) have proposed to compare the activation frequency of convolutional filters in clean and adversarial examples. The authors are suggested to discuss and compare these methods.

(Ma et al., 2018) Characterizing adversarial subspaces using local intrinsic dimensionality, in ICLR 2018.

(Bai et al., 2021] Improving adversarial robustness via channel-wise activation suppressing, in ICLR 2021.

- The definition of the tensor-state distribution $Q_j(c)$ is confusing. It is stated that each element $q_{ij}$ w.r.t. the state $i$ is the value $count_{ij}$ normalized by $n_j$. However, I’m confused about how the $count_{ij}$ is computed. Furthermore, does this definition satisfy that the sum of all probabilities $q_{ij}$ equals 1? If yes, then why $\sum_{i=1}^{n_j}q_{ij}\le 1$? If not, then how can $Q$ be considered and referred to as a PDF?

- The presentation of this paper is poor. First, the authors use graph terms to describe a model, but they do not clarify the definition of nodes and edges. It seems that the authors directly adapt the computation graph constructed by PyTorch. Second, besides the aforementioned problem with $Q$, some notations and subscripts are confusing. The input is sometimes denoted by $x$, sometimes by $\vec x_i$, and sometimes by $\vec x$. Also, the subscript $i$ sometimes represents an input and sometimes represents a tensor state.

- The ``hierarchical’’ in the title seems inappropriate. In my opinion, although the authors consider the neural network as a hierarchical graph, it does not mean that the proposed explanation method is also hierarchical. The authors represent the activation utility of each node (or layer) w.r.t. different samples with a scalar, and such scalars of different nodes are further concatenated to represent the utility of the entire model. Such a representation is not hierarchical.

- The proposed method cannot well address the challenges of high dimension of feature, because this method requires recording the activation values of all computation units over different samples. Furthermore, the computational cost is high. Although the authors mention that the computational cost can be reduced by reducing the number of input samples or limiting the number of probes, these tricks may hurt the accuracy of the obtained explanation.

- About experimental results: (1) In Figure 4, which is a frequency histogram, areas of different models (colors) are supposed to be the same. How do you draw the histogram here? (2) How to understand the raw tensor and unique tensor in Figure 6&7? (3) The datasets and models used in the paper are too limited. The authors are suggested to conduct experiments on various datasets and architectures. (4) The authors do not compare their method with previous methods as baselines.

- Are there any new insights from the results? Although the proposed method indeed shows some differences in clean and poisoned models, how should we understand and then make use of such differences?

**Questions:**

The quality of the figures in this paper is poor.

Some typos: (1) page 4: $i\in\\{1,\ldots,M\\}$ $\to$ $i\in\\{1,\ldots,m\\}$. (2) the third line on page 6: Equation 3 and 4 $\to$ Equation 4 and 5.

---

> ### Author Response · Authors · 2023-11-17
> **response to reviewer  f9Xz**
>
> Weaknesses:
>
> 1.	There have been some.....
>
> Response: Thank you very much for the pointers to the two papers with the two overlapping co-authors: Xingjun Ma and Yisen Wang.
>
> Ma et al., 2018: In this paper, the authors characterize the dimensional properties of adversarial regions in the space of activation values via Local Intrinsic Dimensionality (LID).  This metric defined in Equation (1) of the paper is based on the rate of volume growth with distance. In comparison, our metric has nothing to do with volume growth with distance and is exclusively based on distribution of states of activation values.
>
> Bai et al., (2021): The authors designed a method to suppress redundant activation from being activated by adversarial perturbations via a Channel-wise Activation Suppressing (CAS) strategy. This method is based on two assumptions about adversarial examples from the channel-wise activation perspective: “1) the activation magnitudes of adversarial examples are higher than that of natural examples; and 2) the channels are activated more uniformly by adversarial examples than natural examples.” The key contribution of the paper is captured in Figure 3 of the paper showing the Channel-wise Activation Suppressing (CAS) strategy. In comparison, our paper uses the poisoned TrojAI models that violate these assumptions, presents a metric for measuring utilization of AI model parts, and is focused on understanding the trojan encoding in AI models rather than designing a strategy for suppressing redundant activation like the CAS strategy.
>
> 2. The definition of the tensor-state .....
>
> Response: the tensor-state distribution is a true probability distribution function. Thus, the sum of probabilities equals to one because it is formed by counting the number of occurring activation states divided by the maximum number of possible activation states. To answer the question about counting, here is an example. Given a tensor with dimensions Width, Height, Channels, where the activation values are binary, the count of identical binary vectors along Channel dimension contribute to the histogram bin with the unique binary vector. The reason why this is computationally feasible is that following the cognitive science knowledge and considering nonlinear activation functions like ReLU, the tensor states are binary.
>
> 3. The presentation of this paper ....
>
> Response: (1) It is true that we leveraged the graphs constructed by PyTorch because there is no need to reimplement different implementations of published AI architectures and included in PyTorch. (2) The reason why we needed to have subscripts and include them as needed was the hierarchical nature of our analyses. For example, Equations (1) and (2) use the symbol x with arrow to denote the functional mapping between any input and the output class label. In contrast, the symbol x with arrow and subscript i denotes one particular input from the set of all inputs (training inputs and test inputs).
>
> 4. The ``hierarchical’’ in the title .....
>
> Response: The explanation is declared to be hierarchical because the insights about the trojan encoding is gained at different levels of granularity of the AI architecture. The majority of published work explains Input-Output relationships. Our hierarchical approach provides explanation of the trojan encoding at each layer or at each node.
>
>
> 5. The proposed method cannot well address the challenges.....
>
> Response: It is not necessary to record the activation values of all computation units because they can be analyzed on-the-fly. When we record the binary activation value, we are using the bit storage which is very efficient. It is true that the computational cost depends on the granularity of analyses. We plan to analyze the tradeoffs between accuracy and shortcuts in the future.
>
> 6. About experimental results: .....
>
> Response: (1) Figure 4 illustrates that the utilization of four ResNet101 models with {0,1,2,2} trojans have different utilization values across their AI architecture graph nodes. One can see that the clean (number of trojans = 0) ResNet101 model (blue color) will not have any graph nodes with utilizations that occur when the number of trojans in AI model is larger than zero. (2) Raw tensors are the tensors as produced by each AI architecture graph node. Unique tensors are the tensors found to be unique in the raw tensors. (3) We have conducted many more experiments. However, the page limit does not allow us to include more figures with those results. (4) We would appreciate if the reviewer would suggest a baseline method to compare our utilization based hierarchical approach to explaining trojan encodings.
>
> 7. Are there any new insights from the results? ....
>
> Response: The new insights are in the understanding of the trojan encodings in AI architecture. It is clear from our paper that the class encodings with trojans will clearly depend on the type of trojans, the AI architecture, and the content of training data.

---

> > ### Author Response · Authors · 2023-11-17
> > **response to the questions of reviewer f9Xz**
> >
> > Questions:
> >
> > 1. The quality of the figures in this paper is poor.
> >
> > Response: we can recapture images in figures with a higher resolution setting.  We would appreciate an input on which figures did not convey the information due to their low quality.
> >
> > 2. Some typos: (1) page 4:  (2) the third line on page 6: Equation 3 and 4 Equation 4 and 5.
> >
> > Response: Page 4: lower case m refers to m-variate images. Upper case M refers to the number of pixels. Page 6: we would like to refer to Equations (3) – (5) because all three of them are relevant.

---

### Official Review · Reviewer_dEwM · 2023-11-01

**Soundness:** 2 fair
**Presentation:** 2 fair
**Contribution:** 2 fair
**Rating:** 3
**Confidence:** 4

**Summary:**

This paper attempts to develop an approach for understanding whether an AI model succumbs to poisoning from malicious signals, and if so, what is the extent of the poisoning. The paper adopts an explainable AI perspective by trying to probe trained models through visualizations of feature activations. The experiments are conducted on benchmarks from a challenge from 202 (TrojAI) and reveal differences in what the tensor-state values look like for clean vs. poisoned models.

**Strengths:**

- The paper targets an important problem of identifying poisoned AI Models that can be manipulated by malicious adversaries into producing outputs that have specific failure models.

- The paper is well motivated, and is well positioned in the context of relevant prior works. The related works discussion is thorough and helps in establishing different "high-level" foundational concepts which the paper is based on.

- The approach of analyzing utilization measurements of trained models from their computational graphs is novel in my understanding.

- The paper goes beyond small networks (like LeCun Net) and small datasets like MNIST (common in prior works in this domain), and shows results on real-world traffic datasets. This is helpful in showing that the proposed analysis is potentially scalable to real-world deployment scenarios.

**Weaknesses:**

- The paper is a bit confusing to follow because some technical definitions and low level concepts are not clearly explained / defined. For example, terms like "utilization measure," "class encoding," "poisoned model" need clear definitions since there are crucial for understanding the approach and results.

- The process of poisoning training images seems to induce significant variations in the poisoned vs. clean images. In the real world, such variations are usually minimal. It will be helpful to perform experiments where the variations are not as extreme, and involve more subtle pixel manipulations.

- The paper talks of "AI models" throughout but the experiments are all conducted with ResNet101 models, and are thus limiting in terms of understanding how general is the approach for identification of poisoned models.

- The visualizations in Figs 6 and 7 are interesting, but are not sufficient in terms of understanding the quantitative results clearly. It will be helpful to have more such qualitative visualizations.

- Why are all the experiments only based on traffic sign category identification?

- The approach doesn't have enough details in the paper for replication (for example, training data specs, model specs, visualizations of tensor-state values for random examples) and there is no supplementary material where these details are provided.

**Questions:**

- The paper is a bit confusing to follow because some technical definitions and low level concepts are not clearly explained / defined. For example, terms like "utilization measure," "class encoding," "poisoned model" need clear definitions since there are crucial for understanding the approach and results.

- The process of poisoning training images seems to induce significant variations in the poisoned vs. clean images. In the real world, such variations are usually minimal. It will be helpful to perform experiments where the variations are not as extreme, and involve more subtle pixel manipulations.

- The paper talks of "AI models" throughout but the experiments are all conducted with ResNet101 models, and are thus limiting in terms of understanding how general is the approach for identification of poisoned models.

- The visualizations in Figs 6 and 7 are interesting, but are not sufficient in terms of understanding the quantitative results clearly. It will be helpful to have more such qualitative visualizations.

- Why are all the experiments only based on traffic sign category identification?

- The approach doesn't have enough details in the paper for replication (for example, training data specs, model specs, visualizations of tensor-state values for random examples) and there is no supplementary material where these details are provided.

---

> ### Author Response · Authors · 2023-11-17
> **response to reviewer dEwM**
>
> Weaknesses:
>
> 1.	The paper is a bit confusing to follow because some technical definitions and low level concepts are not clearly explained / defined. For example, terms like "utilization measure," "class encoding," "poisoned model" need clear definitions since there are crucial for understanding the approach and results.
>
> Response: We tried to relate the terms of published concepts.
>
> 	“utilization measure”: ” Following the categorization in the survey on interpreting inner structures of AI models R¨auker et al. (2022), the utilization measurements can be related to concept vectors whose goal is to associate directions in latent space with meaningful concepts. In the works Fong & Vedaldi (2018)(Network 2 Vector) and Bau et al. (2017) (Network Dissection), the distribution of activation maps at each convolutional unit as inputs pass through is used to determine a threshold. Threshold-based segmented activation maps are compared across concepts. In contrast to the previous work Fong & Vedaldi (2018),Bau et al. (2017), our utilization measurements are computed at all computation units in an AI model, the activation maps are binarized at zero, and statistics are computed over a distribution of tensor-states (including the binarized activation maps from convolutional units).”
>
> o	We also tried to define the terms:
>
> 	“utilization measurement”: “The utilization measurements of class encodings are defined by introducing tensor-states measured at the output of each component in AI computation graphs as training data points pass through the AI graph.”
>
> 	“poisoned model": “Training images of traffic signs can be poisoned by adversaries to encode malicious triggers that change trained AI model prediction from a correct traffic sign to another traffic sign in a presence of such a physically realizable trigger (e.g., sticky note or Instagram filter).”
>
> 	“class encoding”: “During the inference of images from the same class, a vector of utilizations over all graph computation units is recorded and denoted as a class encoding. Differences in class encodings can be visualized by color-coded AI computation graphs to contrast class encodings (e.g., clean and poisoned or clean Class A and Class B - see the right side of Figure 1).”
>
> 2.	The process of poisoning training images seems to induce significant variations in the poisoned vs. clean images. In the real world, such variations are usually minimal. It will be helpful to perform experiments where the variations are not as extreme, and involve more subtle pixel manipulations.
>
> Response: Our primary focus was on physically realizable trojans as opposed to arbitrarily small variations. The space of all possible types of trojans is very large and hence we focused on trojans present in physical sensing since they have been of our utmost concern.
>
> 3.	The paper talks of "AI models" throughout but the experiments are all conducted with ResNet101 models, and are thus limiting in terms of understanding how general is the approach for identification of poisoned models.
>
> Response: Our initial paper including other architectures was twice as long as the current and did not meet the ICLR length requirements In the process of trimming, we decide to focus on ResNet101 model as an example to convey the main idea. We were not able to include both, the clarity of the main contribution and the generalizability across many architectures.
>
> 4. The visualizations in Figs 6 and 7 are interesting, but are not sufficient in terms of understanding the quantitative results clearly. It will be helpful to have more such qualitative visualizations.
>
> Response: The quantification of the differences is very much of our interest. We would welcome suggestions for metrics that are beyond traditional distance metrics between two distributions of tensor values.
>
> 5. Why are all the experiments only based on traffic sign category identification?
>
> Response: Our experiments were limited to the available AI models and training data used in the TrojAI challenge. The creation of datasets and AI models is well understood. In addition, our stakeholders from automobile industry are very interested in solving the problem in the context of self-driving cars.
>
> 6. The approach doesn't have enough details in the paper for replication (for example, training data specs, model specs, visualizations of tensor-state values for random examples) and there is no supplementary material where these details are provided.
>
> Response: We will include the pointers to the datasets and trained AI models. For the reviewer, the datasets are directly accessible from https://pages.nist.gov/trojai/docs/data.html#image-based-tasks
> 	image-classification-jun2020
> 	image-classification-aug2020
> 	image-classification-dec2020
> 	image-classification-feb2021

---

> > ### Comment · Reviewer_dEwM · 2023-11-23
> > **response**
> >
> > Dear authors - Thank you for providing responses to the reviews. Unfortunately all my concerns (listed in the weaknesses, and asked for in the questions) still remain. And several of the other reviewers have also raised similar points, which are not adequately explained. In addition, there is no updated pdf that could help clarify some of our concerns. Hence, I am unable to recommend accepting this paper.

---

### Official Review · Reviewer_A5T6 · 2023-11-03

**Soundness:** 3 good
**Presentation:** 2 fair
**Contribution:** 4 excellent
**Rating:** 6
**Confidence:** 3

**Summary:**

This paper aims to uncover patterns in neural network models that under adversarial attacks. They analyzed the patterns from three levels: graph node (computation unit), subgraph, and graph representations. By experiments, they found that clean images and poisoned images may differ significantly in low-level utilizations.

**Strengths:**

1. This paper targets an interesting and important problem, which helps people to understand what actually happen to a neural network under adversarial attack.
2. This paper proposes a novel hierarchical approach to study the inner patterns of neural networks under adversarial attacks.
3. The experiments look convincing and reveal some interesting results.

**Weaknesses:**

1. I find it's a bit hard to follow the details of the paper. The presentation should be improved.
2. I think it is not so surprising that clean images and poisoned images have different utilization patterns, especially when significant perturbations are added to clean images to create poisoned ones. The conclusion of this paper should be more convincing if you use advanced adversarial attack methods, where humans are unable to distiguish the poisoned images.
3. More discussions should be included to explain the significance and the meaning of this work, such as the possible relations to defense strateges.

**Questions:**

1. Does this work provide any useful suggestions in defending against adversatial attacks?

2. There are many works focus on adversarial examples (see Ian Goodfellow's works), which are hardly distinguished by humans. I believe they are more advanced attacks than that used in this paper. Do the conclusions in this paper still hold under those advanced attacks?

3. The experiments show that the patterns of clean and poisoned images could be significantly different when the AI model classifies them as the same class. Such differences should originate from the noise or triggers. What if you just move or rotate a traffic signal in the clean image? Should the patterns change drastically?

---

> ### Author Response · Authors · 2023-11-17
> **response to the reviewer A5T6**
>
> Questions:
>
> 1.	Does this work provide any useful suggestions in defending against adversatial attacks?
>
> Response: one could analyze the correlations between types of trojans and AI architectures to identify the AI graph locations where the trojans are encoded. We have reported results only for two types of physically realizable trojans, such as polygons and Instagram filters available from the TrojAI challenge, and for a subset of typical CV architectures. Defending reliably against any adversarial attack for a well-defined space of architectures would take additional research.
>
> 2.	There are many works focus on adversarial examples (see Ian Goodfellow's works), which are hardly distinguished by humans. I believe they are more advanced attacks than that used in this paper. Do the conclusions in this paper still hold under those advanced attacks?
>
> Response: our research leveraged trained AI models provided by the TrojAI challenge. Those trained AI models are poisoned by only physically realizable trojans. We anticipate that the conclusions would not hold for an arbitrary definition of a trojan (i.e., an advanced attack). Nonetheless, our focus was on trojans present in physical sensing since they are of our utmost concern.
>
> 3.	The experiments show that the patterns of clean and poisoned images could be significantly different when the AI model classifies them as the same class. Such differences should originate from the noise or triggers. What if you just move or rotate a traffic signal in the clean image? Should the patterns change drastically?
>
> Response: we view the image content as a composition of background, foreground important for classification, and physically realizable trigger. Image noise is understood as a random variation of color or brightness information.  Rotation of a traffic sign would be viewed as a variation of the foreground. If the rotated traffic sign is included in the 10,000 training images per class, then the pattern would capture its frequency of occurrence in the encoding. According to hypotheses stated in the paper “Chris Olah, Nick Cammarata, Ludwig Schubert, Gabriel Goh, Michael Petrov, and Shan Carter.Zoom in: An introduction to circuits. Distill, 2020. doi: 10.23915/distill.00024.001. https://distill.pub/2020/circuits/zoom-in.”, certain shapes and orientations would be encoded in subgraphs and hence adding examples of rotated traffic sign to a training data set would change the pattern.

---

### Meta-Review · Area_Chair_KTGB · 2023-12-07

**Metareview:**

This paper introduces an analysis of AI model poisoning by looking at AI model utilisation at different levels of granularity.

Unfortunately, the reviewers struggled to understand the contributions and novelty of the paper. leading to a near universal initial "reject" which did not get turned around in the limited discussion.

Based on my own read of the paper, the results and approach seem interesting, but I also agree with the reviewers that the paper is difficult to follow. For example, it would be good to explain terms like "hidden classes" and "utilization-based class encodings" (and why they are good) in the introduction.

I believe that the authors could strengthen the paper by better describing their motivation and main take aways from the work.

**Justification For Why Not Higher Score:**

Some of the paper is currently difficult to follow which makes it difficult to assess the novelty and relevance of the work.

**Justification For Why Not Lower Score:**

N/A

---

### Decision · Program_Chairs · 2024-01-16

Reject